# Boosting Adversarial Training with Hypersphere Embedding

**Tianyu Pang**[*], **Xiao Yang**[*]**, Yinpeng Dong, Kun Xu, Jun Zhu, Hang Su**[†]
Dept. of Comp. Sci. & Tech., Institute for AI, BNRist Center
Tsinghua-Bosch Joint ML Center, THBI Lab, Tsinghua University, Beijing, China
{pty17, yangxiao19, dyp17}@mails.tsinghua.edu.cn
kunxu.thu@gmail.com, {suhangss, dcszj}@mail.tsinghua.edu.cn

## Abstract

Adversarial training (AT) is one of the most effective defenses against adversarial attacks for deep learning models. In this work, we advocate incorporating the hypersphere embedding (HE) mechanism into the AT procedure by regularizing the features onto compact manifolds, which constitutes a lightweight yet effective module to blend in the strength of representation learning. Our extensive analyses reveal that AT and HE are well coupled to benefit the robustness of the adversarially trained models from several aspects. We validate the effectiveness and adaptability of HE by embedding it into the popular AT frameworks including PGD-AT, ALP, and TRADES, as well as the FreeAT and FastAT strategies. In the experiments, we evaluate our methods under a wide range of adversarial attacks on the CIFAR-10 and ImageNet datasets, which verifies that integrating HE can consistently enhance the model robustness for each AT framework with little extra computation.

## 1   Introduction

The adversarial vulnerability of deep learning models has been widely recognized in recent years [4, 22, 60]. To mitigate this potential threat, a number of defenses have been proposed, but most of them are ultimately defeated by the attacks adapted to the specific details of the defenses [2, 8]. Among the existing defenses, **adversarial training (AT)** is a general strategy achieving the state-of-the-art robustness under different settings [53, 63, 71, 73, 74, 76, 83]. Various efforts have been devoted to improving AT from different aspects, including accelerating the training procedure [56, 57, 70, 79] and exploiting extra labeled and unlabeled training data [1, 10, 25, 78], which are conducive in the cases with limited computational resources or additional data accessibility.

In the meanwhile, another research route focuses on boosting the adversarially trained models via imposing more direct supervision to regularize the learned representations. Along this line, recent progress shows that encoding triplet-wise metric learning or maximizing the optimal transport (OT) distance of data batch in AT is effective to leverage the inter-sample interactions, which can promote the learning of robust classifiers [34, 41, 45, 80]. However, optimization on the sampled triplets or the OT distance is usually of high computational cost, while the sampling process in metric learning could also introduce extra class biases on unbalanced data [48, 55].

In this work, we provide a lightweight yet competent module to tackle several defects in the learning dynamics of existing AT frameworks, and facilitate the adversarially trained networks learning more robust features. Methodologically, we augment the AT frameworks by integrating the **hypersphere embedding (HE)** mechanism, which normalizes the features in the penultimate layer and the weights in the softmax layer with an additive angular margin. Except for the generic benefits of HE on

---

[*]Equal contribution. [†] Corresponding author.

Table 1: Formulations of the AT frameworks without (✘) or with (✔) HE. The notations are defined in Sec. 2.2. We substitute the adversarial attacks in ALP with untargeted PGD as suggested [18].

| Strategy | HE | Training objective $\mathcal{L}_{\text{T}}$ | Adversarial objective $\mathcal{L}_{\text{A}}$ |
|---|---|---|---|
| PGD-AT | ✘ | $\mathcal{L}_{\text{CE}}(f(x^*), y)$ | $\mathcal{L}_{\text{CE}}(f(x'), y)$ |
| | ✔ | $\mathcal{L}_{\text{CE}}^m(\widetilde{f}(x^*), y)$ | $\mathcal{L}_{\text{CE}}(\widetilde{f}(x'), y)$ |
| ALP | ✘ | $\alpha\,\mathcal{L}_{\text{CE}}(f(x), y) + (1-\alpha)\,\mathcal{L}_{\text{CE}}(f(x^*), y) + \lambda\|\mathbf{W}^\top(z-z^*)\|_2$ | $\mathcal{L}_{\text{CE}}(f(x'), y)$ |
| | ✔ | $\alpha\,\mathcal{L}_{\text{CE}}^m(\widetilde{f}(x), y) + (1-\alpha)\,\mathcal{L}_{\text{CE}}^m(\widetilde{f}(x^*), y) + \lambda\|\widetilde{\mathbf{W}}^\top(\widetilde{z}-\widetilde{z}^*)\|_2$ | $\mathcal{L}_{\text{CE}}(\widetilde{f}(x'), y)$ |
| TRADES | ✘ | $\mathcal{L}_{\text{CE}}(f(x), y) + \lambda\,\mathcal{L}_{\text{CE}}(f(x^*), f(x))$ | $\mathcal{L}_{\text{CE}}(f(x'), f(x))$ |
| | ✔ | $\mathcal{L}_{\text{CE}}^m(\widetilde{f}(x), y) + \lambda\,\mathcal{L}_{\text{CE}}(\widetilde{f}(x^*), \widetilde{f}(x))$ | $\mathcal{L}_{\text{CE}}(\widetilde{f}(x'), \widetilde{f}(x))$ |

learning angularly discriminative representations [37, 39, 67, 72], we contribute to the extensive analyses (detailed in Sec. 3) showing that the encoded HE mechanism naturally adapts to AT.

To intuitively explain the main insights, we take a binary classification task as an example, where the cross-entropy (CE) objective equals to maximizing $\mathcal{L}(x) = (W_0 - W_1)^\top z = \|W_{01}\|\|z\|\cos(\theta)$ on an input $x$ with label $y = 0$. (i) If $x$ is correctly classified, there is $\mathcal{L}(x) > 0$, and adversaries aim to craft $x'$ such that $\mathcal{L}(x') < 0$. Since $\|W_{01}\|$ and $\|z\|$ are always positive, they cannot alter the sign of $\mathcal{L}$. Thus feature normalization (FN) and weight normalization (WN) encourage the adversaries to attack the crucial component $\cos(\theta)$, which results in more efficient perturbations when crafting adversarial examples in AT; (ii) In a data batch, points with larger $\|z\|$ will dominate (vicious circle on increasing $\|z\|$), which makes the model ignore the critical component $\cos(\theta)$. FN alleviates this problem by encouraging the model to devote more efforts on learning hard examples, and well-learned hard examples will dynamically have smaller weights during training since $\cos(\theta)$ is bounded; This can promote the worst-case performance under adversarial attacks; (iii) When there are much more samples of label 0, the CE objective will tend to have $\|W_0\| \gg \|W_1\|$ to minimize the loss. WN can relieve this trend and encourage $W_0$ and $W_1$ to diversify in directions. This mechanism alleviates the unbalanced label distributions caused by the untargeted or multi-targeted attacks applied in AT [23, 40], where the resulted adversarial labels depend on the semantic similarity among classes; (iv) The angular margin (AM) induces a larger inter-class variance and margin under the angular metric to further improve model robustness, which plays a similar role as the margin in SVM.

Our method is concise and easy to implement. To validate the effectiveness, we consider three typical AT frameworks to incorporate with HE, namely, **PGD-AT** [40], **ALP** [29], and **TRADES** [81], as summarized in Table 1. We further verify the generality of our method by evaluating the combination of HE with previous strategies on accelerating AT, e.g., **FreeAT** [57] and **FastAT** [70]. In Sec. 4, we empirically evaluate the defenses on CIFAR-10 [31] and ImageNet [15] under several different adversarial attacks, including the commonly adopted PGD [40] and other strong ones like the feature attack [35], FAB [13], SPSA [65], and NES [27], etc. We also test on the CIFAR-10-C and ImageNet-C datasets with corrupted images to inspect the robustness under general transformations [24]. The results demonstrate that incorporating HE can consistently improve the performance of the models trained by each AT framework, while introducing little extra computation.

## 2 Methodology

In this section, we define the notations, introduce the hypersphere embedding (HE) mechanism, and provide the formulations under the adversarial training (AT) frameworks. Due to the limited space, we extensively introduce the related work in Appendix B, including those on combining metric learning with AT [34, 41, 44, 46, 80] and further present their bottlenecks.

### 2.1 Notations

For the classification task with $L$ labels in $[L] := \{1, \cdots, L\}$, a deep neural network (DNN) can be generally denoted as the mapping function $f(x)$ for the input $x$ as

$$f(x) = \mathbb{S}(\mathbf{W}^\top z + b), \tag{1}$$

where $z = z(x; \boldsymbol{\omega})$ is the extracted feature with model parameters $\boldsymbol{\omega}$, the matrix $\mathbf{W} = (W_1, \cdots, W_L)$ and vector $b$ are respectively the weight and bias in the softmax layer, and $\mathbb{S}(h) : \mathbb{R}^L \to \mathbb{R}^L$ is the

softmax function. One common training objective for DNNs is the cross-entropy (CE) loss defined as

$$\mathcal{L}_{\text{CE}}(f(x), y) = -1_y^\top \log f(x), \tag{2}$$

where $1_y$ is the one-hot encoding of label $y$ and the logarithm of a vector is taken element-wisely. In this paper, we use $\angle(u, v)$ to denote the angle between vectors $u$ and $v$.

## 2.2 The AT frameworks with HE

Adversarial training (AT) is one of the most effective and widely studied defense strategies against adversarial vulnerability [6, 33]. Most of the AT methods can be formulated as a two-stage framework:

$$\min_{\boldsymbol{\omega}, \mathbf{W}} \mathbb{E}\left[\mathcal{L}_{\text{T}}(\boldsymbol{\omega}, \mathbf{W}|x, x^*, y)\right], \text{ where } x^* = \arg\max_{x' \in \mathbf{B}(x)} \mathcal{L}_{\text{A}}(x'|x, y, \boldsymbol{\omega}, \mathbf{W}). \tag{3}$$

Here $\mathbb{E}[\cdot]$ is the expectation w.r.t. the data distribution, $\mathbf{B}(x)$ is a set of allowed points around $x$, $\mathcal{L}_{\text{T}}$ and $\mathcal{L}_{\text{A}}$ are the training and adversarial objectives, respectively. Since the inner maximization and outer minimization problems are mutually coupled, they are iteratively executed in training until the model parameters $\boldsymbol{\omega}$ and $\mathbf{W}$ converge [40]. To promote the performance of the adversarially trained models, recent work proposes to embed pair-wise or triplet-wise metric learning into AT [34, 41, 80], which facilitates the neural networks learning more robust representations. Although these methods are appealing, they could introduce high computational overhead [41], cause unexpected class biases [26], or be vulnerable under strong adversarial attacks [35].

In this paper, we address the above deficiencies by presenting a lightweight yet effective module that integrates the **hypersphere embedding** (**HE**) mechanism with an AT procedure. Though HE is not completely new, our analysis in Sec. 3 demonstrates that HE naturally adapts to the learning dynamics of AT and can induce several advantages special to the adversarial setting. Specifically, the HE mechanism involves three typical operations including feature normalization (FN), weight normalization (WN), and angular margins (AM), as described below.

Note that in Eq. (1) there is $\mathbf{W}^\top z = (W_1^\top z, \cdots, W_L^\top z)$, and $\forall l \in [L]$, the inner product $W_l^\top z = \|W_l\|\|z\|\cos(\theta_l)$, where $\theta_l = \angle(W_l, z)$.[1] Then the WN and FN operations can be denoted as

$$\textbf{WN operation: } \widetilde{W_l} = \frac{W_l}{\|W_l\|}; \textbf{ FN operation: } \widetilde{z} = \frac{z}{\|z\|}, \tag{4}$$

Let $\cos\boldsymbol{\theta} = (\cos(\theta_1), \cdots, \cos(\theta_L))$ and $\widetilde{\mathbf{W}}$ be the weight matrix after executing WN on each column vector $W_l$. Then, the output predictions of the DNNs with HE become

$$\widetilde{f}(x) = \mathbb{S}(\widetilde{\mathbf{W}}^\top \widetilde{z}) = \mathbb{S}(\cos\boldsymbol{\theta}), \tag{5}$$

where no bias vector $b$ exists in $\widetilde{f}(x)$ [38, 67]. In contrast, the **AM operation** is only performed in the training phase, where $\widetilde{f}(x)$ is fed into the CE loss with a margin $m$ [68], formulated as

$$\mathcal{L}_{\text{CE}}^m(\widetilde{f}(x), y) = -1_y^\top \log \mathbb{S}(s \cdot (\cos\boldsymbol{\theta} - m \cdot 1_y)). \tag{6}$$

Here $s > 0$ is a hyperparameter to improve the numerical stability during training [67]. To highlight our main contributions in terms of methodology, we summarize the proposed formulas of AT in Table 1. We mainly consider three representative AT frameworks including **PGD-AT** [40], **ALP** [29], and **TRADES** [81]. The differences between our enhanced versions (with HE) from the original versions (without HE) are colorized. Note that we apply the HE mechanism both on the adversarial objective $\mathcal{L}_{\text{A}}$ for constructing adversarial examples (the inner maximization problem), and the training objective $\mathcal{L}_{\text{T}}$ for updating parameters (the outer minimization problem).

## 3 Analysis of the benefits

In this section, we analyze the benefits induced by the mutual interaction between AT and HE under the $\ell_p$-bounded threat model [9], where $\mathbf{B}(x) = \{x' | \|x' - x\|_p \leq \epsilon\}$ and $\epsilon$ is the maximal perturbation. Detailed proofs for the conclusions below can be found in Appendix A.

## 3.1 Formalized first-order adversary

Most of the adversarial attacks applied in AT belong to the family of first-order adversaries [58], due to the computational efficiency. We first define the vector function $\mathbb{U}_p$ as

$$\mathbb{U}_p(u) = \underset{\|v\|_p \leq 1}{\arg\max} \, u^\top v, \text{ where } u^\top \mathbb{U}_p(u) = \|u\|_q. \tag{7}$$

Here $\|\cdot\|_q$ is the dual norm of $\|\cdot\|_p$ with $\frac{1}{p} + \frac{1}{q} = 1$ [5]. Specially, there are $\mathbb{U}_2(u) = \frac{u}{\|u\|_2}$ and $\mathbb{U}_\infty(u) = \text{sign}(u)$. If $u = \nabla_x \mathcal{L}_A$, then $\mathbb{U}_p(\nabla_x \mathcal{L}_A)$ is the direction of greatest increase of $\mathcal{L}_A$ under the first-order Taylor's expansion [30] and the $\ell_p$-norm constraint, as stated below:

**Lemma 1.** *(First-order adversary) Given the adversarial objective $\mathcal{L}_A$ and the set $\mathbf{B}(x) = \{x' | \|x' - x\|_p \leq \epsilon\}$, under the first-order Taylor's expansion, the solution for $\max_{x' \in B(x)} \mathcal{L}_A(x')$ is $x^* = x + \epsilon \mathbb{U}_p(\nabla_x \mathcal{L}_A(x))$. Furthermore, there is $\mathcal{L}_A(x^*) = \mathcal{L}_A(x) + \epsilon \|\nabla_x \mathcal{L}_A(x)\|_q$.*

According to the one-step formula in Lemma 1, we can generalize to the multi-step generation process of first-order adversaries under the $\ell_p$-bounded threat model. For example, in the $t$-th step of the iterative attack with step size $\eta$ [32], the adversarial example $x^{(t)}$ is updated as

$$x^{(t)} = x^{(t-1)} + \eta \mathbb{U}_p(\nabla_x \mathcal{L}_A(x^{(t-1)})), \tag{8}$$

where the increment of the loss is $\Delta \mathcal{L}_A = \mathcal{L}_A(x^{(t)}) - \mathcal{L}_A(x^{(t-1)}) = \eta \|\nabla_x \mathcal{L}_A(x^{(t-1)})\|_q$.

## 3.2 The inner maximization problem in AT

As shown in Table 1, the adversarial objectives $\mathcal{L}_A$ are usually the CE loss between the adversarial prediction $f(x')$ and the target prediction $f(x)$ or $1_y$. Thus, to investigate the inner maximization problem of $\mathcal{L}_A$, we expand the gradient of CE loss w.r.t. $x'$ as below:

**Lemma 2.** *(The gradient of CE loss) Let $W_{ij} = W_i - W_j$ be the residual vector between two weights, and $z' = z(x'; \boldsymbol{\omega})$ be the mapped feature of the adversarial example $x'$, then there is*

$$\nabla_{x'} \mathcal{L}_{CE}(f(x'), f(x)) = -\sum_{i \neq j} f(x)_i f(x')_j \nabla_{x'}(W_{ij}^\top z'). \tag{9}$$

*If $f(x) = 1_y$ is the one-hot label vector, we have $\nabla_{x'} \mathcal{L}_{CE}(f(x'), y) = -\sum_{l \neq y} f(x')_l \nabla_{x'}(W_{yl}^\top z')$.*

Lemma 2 indicates that the gradient of CE loss can be decomposed into the linear combination of the gradients on the residual logits $W_{ij}^\top z'$. Let $y^*$ be the predicted label on the finally crafted adversarial example $x^*$, where $y^* \neq y$. Based on the empirical observations [21, 43], we are justified to assume that $f(x)_y$ is much larger than $f(x)_l$ for $l \neq y$, and $f(x')_{y^*}$ is much larger than $f(x')_l$ for $l \notin \{y, y^*\}$. Then we can approximate the linear combination in Eq. (9) with the dominated term as

$$\nabla_{x'} \mathcal{L}_{CE}(f(x'), f(x)) \approx -f(x)_y f(x')_{y^*} \nabla_{x'}(W_{yy^*}^\top z'), \text{ where } W_{yy^*} = W_y - W_{y^*}. \tag{10}$$

Let $\theta'_{yy^*} = \angle(W_{yy^*}, z')$, there is $W_{yy^*}^\top z' = \|W_{yy^*}\| \|z'\| \cos(\theta'_{yy^*})$ and $W_{yy^*}$ does not depend on $x'$. Thus by substituting Eq. (10) into Eq. (8), the update direction of each attacking step becomes

$$\mathbb{U}_p[\nabla_{x'} \mathcal{L}_{CE}(f(x'), f(x))] \approx -\mathbb{U}_p[\nabla_{x'}(\|z'\| \cos(\theta'_{yy^*}))], \tag{11}$$

where the factor $f(x)_y f(x')_{y^*}$ is eliminated according to the definition of $\mathbb{U}_p[\cdot]$. Note that Eq. (11) also holds when $f(x) = 1_y$, and the resulted adversarial objective is analogous to the C&W attack [7].

## 3.3 Benefits from feature normalization

To investigate the effects of FN alone, we deactivate the WN operation in Eq. (5) and denote

$$\overline{f}(x) = \mathbb{S}(\mathbf{W}^\top \widetilde{z}). \tag{12}$$

Then similar to Eq. (11), we can obtain the update direction of the attack with FN applied as

$$\mathbb{U}_p[\nabla_{x'} \mathcal{L}_{CE}(\overline{f}(x'), \overline{f}(x))] \approx -\mathbb{U}_p[\nabla_{x'}(\cos(\theta'_{yy^*}))]. \tag{13}$$

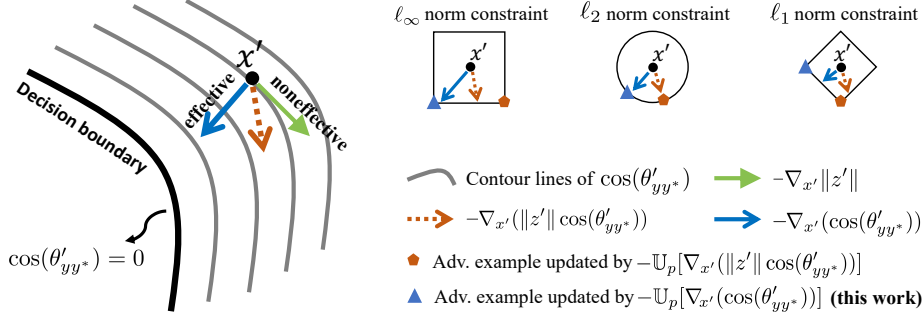

Figure 1: Intuitive illustration in the input space. When applying FN in $\mathcal{L}_{\text{A}}$, the adversary can take more effective update steps to move $x'$ across the decision boundary defined by $\cos(\theta'_{yy^*}) = 0$.

**More effective adversarial perturbations.** In the AT procedure, we prefer to craft adversarial examples more efficiently to reduce the computational burden [36, 66, 70]. As shown in Fig. 1, to successfully fool the model to classify $x'$ into the label $y^*$, the adversary needs to craft iterative perturbations to move $x'$ across the decision boundary defined by $\cos(\theta'_{yy^*}) = 0$. Under the first-order optimization, the most effective direction to reach the decision boundary is along $-\nabla_{x'}(\cos(\theta'_{yy^*}))$, namely, the direction with the steepest descent of $\cos(\theta'_{yy^*})$. In contrast, the direction of $-\nabla_{x'}\|z'\|$ is nearly tangent to the contours of $\cos(\theta'_{yy^*})$, especially in high-dimension spaces, which is noneffective as the adversarial perturbation. Actually from Eq. (1) we can observe that when there is no bias term in the softmax layer, changing the norm $\|z'\|$ will not affect the predicted labels at all.

By comparing Eq. (11) and Eq. (13), we can find that applying FN in the adversarial objective $\mathcal{L}_{\text{A}}$ exactly removes the noneffective component caused by $\|z'\|$, and encourages the adversarial perturbations to be aligned with the effective direction $-\nabla_{x'}(\cos(\theta'_{yy^*}))$ under the $\ell_p$-norm constraint. This facilitate crafting adversarial examples with fewer iterations and improve the efficiency of the AT progress, as empirically verified in the left panel of Fig. 2. Besides, in Fig. 1 we also provide three instantiations of the $\ell_p$-norm constraint. We can see that if we do not use FN, the impact of the noneffective component of $-\nabla_{x'}\|z'\|$ could be magnified under, e.g., the $\ell_\infty$-norm constraint, which could consequently require more iterative steps and degrade the training efficiency.

**Better learning on hard (adversarial) examples.** As to the benefits of applying FN in the training objective $\mathcal{L}_{\text{T}}$, we formally show that FN can promote learning on hard examples, as empirically observed in the previous work [50]. In the adversarial setting, this property can promote the worst-case performance under potential adversarial threats. Specifically, the model parameters $\boldsymbol{\omega}$ is updated towards $-\nabla_{\boldsymbol{\omega}} \mathcal{L}_{\text{CE}}$. When FN is not applied, we can use similar derivations as in Lemma 2 to obtain

$$-\nabla_{\boldsymbol{\omega}} \mathcal{L}_{\text{CE}}(f(x), y) = \sum_{l \neq y} f(x)_l \|W_{yl}\|(\cos(\theta_{yl})\nabla_{\boldsymbol{\omega}}\|z\| + \|z\|\nabla_{\boldsymbol{\omega}}\cos(\theta_{yl})), \qquad (14)$$

where $W_{yl} = W_y - W_l$ and $\theta_{yl} = \angle(W_{yl}, z)$. According to Eq. (14), when we use a mini-batch of data to update $\boldsymbol{\omega}$, the inputs with small $\nabla_{\boldsymbol{\omega}}\|z\|$ or $\nabla_{\boldsymbol{\omega}}\cos(\theta_{yl})$ contribute less in the direction of model updating, which are qualitatively regarded as hard examples [50]. This causes the training process to devote noneffective efforts to increasing $\|z\|$ for easy examples and consequently overlook the hard ones, which leads to vicious circles and could degrade the model robustness against strong adversarial attacks [51, 59]. As shown in Fig. 3, the hard examples in AT are usually the crafted adversarial examples, which are those we actually expect the model to focus on in the AT procedure. In comparison, when FN is applied, there is $\nabla_{\boldsymbol{\omega}}\|\widetilde{z}\| = 0$, then $\boldsymbol{\omega}$ is updated towards

$$-\nabla_{\boldsymbol{\omega}} \mathcal{L}_{\text{CE}}(\overline{f}(x), y) = \sum_{l \neq y} \overline{f}(x)_l \|W_{yl}\|\nabla_{\boldsymbol{\omega}}\cos(\theta_{yl}). \qquad (15)$$

In this case, due to the bounded value range $[-1, 1]$ of cosine function, the easy examples will contribute less when they are well learned, i.e., have large $\cos(\theta_{yl})$, while the hard examples could later dominate the training. This causes a dynamic training procedure similar to curriculum learning [3].

### 3.4 Benefits from weight normalization

In the AT procedure, we usually apply untargeted attacks [18, 40]. Since we do not explicit assign targets, the resulted prediction labels and feature locations of the crafted adversarial examples will

Table 2: Classification accuracy (%) on **CIFAR-10** under the *white-box* threat model. The perturbation $\epsilon = 0.031$, step size $\eta = 0.003$. We highlight the best-performance model under each attack.

| Defense | Clean | PGD-20 | PGD-500 | MIM-20 | FGSM | DeepFool | C&W | FeaAtt. | FAB |
|---------|-------|--------|---------|--------|------|----------|-----|---------|-----|
| PGD-AT | 86.75 | 53.97 | 51.63 | 55.08 | 59.70 | 57.26 | 84.00 | 52.38 | 51.23 |
| PGD-AT+**HE** | 86.19 | 59.36 | 57.59 | 60.19 | 63.77 | 61.56 | 84.07 | 52.88 | **54.45** |
| ALP | 87.18 | 52.29 | 50.13 | 53.35 | 58.99 | 59.40 | 84.96 | 49.55 | 50.54 |
| ALP+**HE** | **89.91** | 57.69 | 51.78 | 58.63 | 65.08 | **65.19** | **87.86** | 48.64 | 51.86 |
| TRADES | 84.62 | 56.48 | 54.84 | 57.14 | 61.02 | 60.70 | 81.13 | 55.09 | 53.58 |
| TRADES+**HE** | 84.88 | **62.02** | **60.75** | **62.71** | **65.69** | 60.48 | 81.44 | **58.13** | 53.50 |

Table 3: Validation of combining FastAT and FreeAT with HE and m-HE on **CIFAR-10**. We report the accuracy (%) on clean and PGD, as well as the total training time (min).

| Defense | Epo. | Clean | PGD-50 | Time |
|---------|------|-------|--------|------|
| FastAT | 30 | **83.80** | 46.40 | *11.38* |
| FastAT+**HE** | 30 | 82.58 | 52.55 | *11.48* |
| FastAT+**m-HE** | 30 | 83.14 | **53.49** | *11.49* |
| FreeAT | 10 | 77.21 | 46.14 | *15.78* |
| FreeAT+**HE** | 10 | 76.85 | 50.98 | *15.87* |
| FreeAT+**m-HE** | 10 | **77.59** | **51.85** | *15.91* |

Table 4: Top-1 classification accuracy (%) on **ImageNet** under the *white-box* threat model.

| Model | Method | Clean | PGD-10 | PGD-50 |
|-------|--------|-------|--------|--------|
| ResNet-50 | FreeAT | 60.28 | 32.13 | 31.39 |
| | FreeAT+**HE** | **61.83** | **40.22** | **39.85** |
| ResNet-152 | FreeAT | 65.20 | 36.97 | 35.87 |
| | FreeAT+**HE** | **65.41** | **43.24** | **42.60** |
| WRN-50-2 | FreeAT | 64.18 | 36.24 | 35.38 |
| | FreeAT+**HE** | **65.28** | **43.83** | **43.47** |
| WRN-101-2 | FreeAT | 66.15 | 39.35 | 38.23 |
| | FreeAT+**HE** | **66.37** | **45.35** | **45.04** |

depend on the unbalanced semantic similarity among different classes. For example, the learned features of dogs and cats are usually closer than those between dogs and planes, so an untargeted adversary will prefer to fool the model to predict the cat label on a dog image, rather than the plane label [42]. To understand how the adversarial class biases affect training, assuming that we perform the gradient descent on a data batch $\mathcal{D} = \{(x^k, y^k)\}_{k \in [N]}$. Then we can derive that $\forall l \in [L]$, the softmax weight $W_l$ is updated towards

$$- \nabla_{W_l} \mathcal{L}_{\text{CE}}(\mathcal{D}) = \sum_{x^k \in \mathcal{D}_l} z^k - \sum_{x^k \in \mathcal{D}} f(x^k)_l \cdot z^k, \tag{16}$$

where $\mathcal{D}_l$ is the subset of $\mathcal{D}$ with true label $l$. We can see that the weight $W_l$ will tend to have larger norm when there are more data or easy examples in class $l$, i.e., larger $|\mathcal{D}_l|$ or $\|z^k\|$ for $x^k \in \mathcal{D}_l$. Besides, if an input $x$ in the batch is adversarial, then $f(x)_y$ is usually small and consequently $z$ will have a large effect on the update of $W_y$. Since there is $W_{yy^*}^\top z < 0$, $W_y$ will be updated towards $W_{y^*}$ both in norm and direction, which causes repetitive oscillation during training.

When applying WN, the update of $W_l$ will only depend on the averaged feature direction within each class, which alleviates the noneffective oscillation on the weight norm and speed up training [52]. Besides, when FN and WN are both applied, the inner products $\boldsymbol{W}^\top z$ in the softmax layer will become the angular metric $\cos \boldsymbol{\theta}$, as shown in Eq. (5). Then we can naturally introduce AM to learn angularly more discriminative and robust features [17, 75].

### 3.5 Modifications to better utilize strong adversaries

In most of the AT procedures, the crafted adversarial examples will only be used once in a single training step to update the model parameters [40], which means hard adversarial examples may not have an chance to gradually dominate the training as introduced in Eq. (15). Since $\nabla_{\boldsymbol{\omega}} \cos(\theta_{yy^*}) = -\sin(\theta_{yy^*}) \nabla_{\boldsymbol{\omega}} \theta_{yy^*}$, the weak adversarial examples around the decision boundary with $\theta_{yy^*} \sim 90°$ have higher weights $\sin(\theta_{yy^*})$. This makes the model tend to overlook the strong adversarial examples with large $\theta_{yy^*}$, which contain abundant information. To be better compatible with strong adversaries, an easy-to-implement way is to directly substitute $\mathbb{S}(\cos \boldsymbol{\theta})$ in Eq. (5) with $\mathbb{S}(-\boldsymbol{\theta})$, using the $\arccos$ operator. We name this form of embedding as modified HE (**m-HE**), as evaluated in Table 3.

Table 5: Top-1 classification accuracy (%) on **CIFAR-10-C** and **ImageNet-C**. The models are trained on the original datasets CIFAR-10 and ImageNet, respectively. Here 'mCA' refers to the mean accuracy averaged on different corruptions and severity. Full version of the table is in Appendix C.6.

| Defense | mCA | Blur | | | | Weather | | | | Digital | | | |
|---|---|---|---|---|---|---|---|---|---|---|---|---|---|
| | | Defocus | Glass | Motion | Zoom | Snow | Frost | Fog | Bright | Contra | Elastic | Pixel | JPEG |
| **CIFAR-10-C** | | | | | | | | | | | | | |
| PGD-AT | 77.23 | 81.84 | 79.69 | 77.62 | 80.88 | 81.32 | 77.95 | 61.70 | 84.05 | 44.55 | 80.79 | 84.76 | 84.35 |
| PGD-AT+**HE** | **77.29** | 81.86 | 79.45 | 78.17 | 80.87 | 80.77 | 77.98 | 62.45 | 83.67 | 45.11 | 80.69 | 84.16 | 84.10 |
| ALP | 77.73 | 81.94 | 80.31 | 78.23 | 80.97 | 81.74 | 79.26 | 61.51 | 84.88 | 45.86 | 80.91 | 85.09 | 84.68 |
| ALP+**HE** | **80.55** | 80.87 | 85.23 | 81.26 | 84.43 | 85.14 | 83.89 | 68.83 | 88.33 | 50.74 | 84.44 | 87.44 | 87.28 |
| TRADES | 75.36 | 79.84 | 77.72 | 76.34 | 78.66 | 79.52 | 76.94 | 59.68 | 82.06 | 43.80 | 78.53 | 82.65 | 82.31 |
| TRADES+**HE** | **75.78** | 80.55 | 77.61 | 77.26 | 79.62 | 79.23 | 76.53 | 61.39 | 82.33 | 45.04 | 79.29 | 82.50 | 82.40 |
| **ImageNet-C** | | | | | | | | | | | | | |
| FreeAT | 28.22 | 19.15 | 26.63 | 25.75 | 28.25 | 23.03 | 23.47 | 3.71 | 45.18 | 5.40 | 41.76 | 48.78 | 52.55 |
| FreeAT+**HE** | **30.04** | 21.16 | 29.28 | 28.08 | 30.76 | 26.62 | 28.35 | 5.34 | 49.88 | 7.03 | 44.72 | 51.17 | 55.05 |

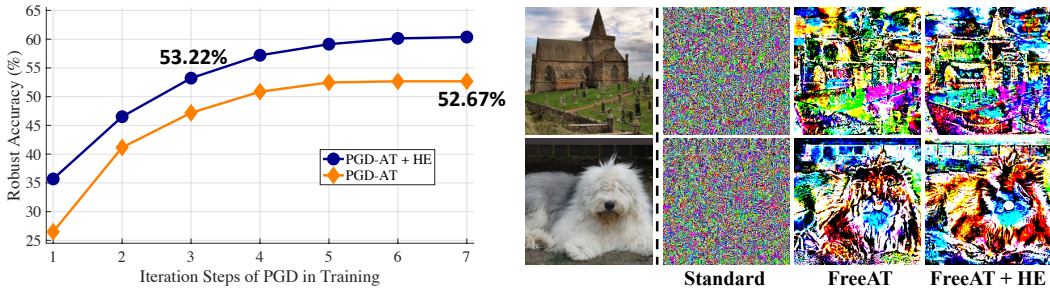

Figure 2: *Left.* Accuracy (%) under PGD-20, where the models are trained by PGD-AT with different iteration steps on **CIFAR-10**. *Right.* Visualization of the adversarial perturbations on **ImageNet**.

## 4 Experiments

**CIFAR-10 [31] setup.** We apply the wide residual network WRN-34-10 as the model architecture [77]. For each AT framework, we set the maximal perturbation $\epsilon = 8/255$, the perturbation step size $\eta = 2/255$, and the number of iterations $K = 10$. We apply the momentum SGD [49] optimizer with the initial learning rate of $0.1$, and train for 100 epochs. The learning rate decays with a factor of $0.1$ at 75 and 90 epochs, respectively. The mini-batch size is 128. Besides, we set the regularization parameter $1/\lambda$ as 6 for TRADES, and set the adversarial logit pairing weight as 0.5 for ALP [29, 81]. The scale $s = 15$ and the margin $m = 0.2$ in HE, where different $s$ and $m$ correspond to different trade-offs between the accuracy and robustness, as detailed in Appendix C.3.

**ImageNet [15] setup.** We apply the framework of free adversarial training (**FreeAT**) in Shafahi et al. [57], which has a similar training objective as PGD-AT and can train a robust model using four GPU workers. We set the repeat times $m = 4$ in FreeAT. The perturbation $\epsilon = 4/255$ with the step size $\eta = 1/255$. We train the model for 90 epochs with the initial learning rate of $0.1$, and the mini-batch size is 256. The scale $s = 10$ and the margin $m = 0.2$ in HE.[2]

### 4.1 Performance under white-box attacks

On the CIFAR-10 dataset, we test the defenses under different attacks including FGSM [22], PGD [40], MIM [16], Deepfool [42], C&W ($l_\infty$ version) [7], feature attack [35], and FAB [13]. We report the classification accuracy in Table 2 following the evaluation settings in Zhang et al. [81]. We denote the iteration steps behind the attacking method, e.g., 10-step PGD as PGD-10. To verify that our strategy is generally compatible with previous work on accelerating AT, we combine HE with the one-step based FreeAT and fast adversarial training (**FastAT**) frameworks [70]. We provide the accuracy and training time results in Table 3. We can see that the operations in HE increase negligible computation, even in the cases pursuing extremely fast training. Besides, we also evaluate embedding **m-HE** (introduced in Sec. 3.5) and find it more effective than HE when combining with PGD-AT, FreeAT and Fast AT that exclusively train on adversarial examples. On the ImageNet dataset, we follow the evaluation settings in Shafahi et al. [57] to test under PGD-10 and PGD-50, as shown in Table 4.

Table 6: Classification accuracy (%) on the clean test data, and under two benchmark attacks RayS and AutoAttack.

| Method | Architecture | Clean | RayS | AA |
|---|---|---|---|---|
| PGD-AT+**HE** | WRN-34-10 | 86.25 | 57.8 | 53.16 |
| | WRN-34-20 | 85.14 | 59.0 | 53.74 |

Table 7: Attacking standardly trained WRN-34-10 with or without FN.

| Attack | FN | Acc. (%) |
|---|---|---|
| PGD-1 | ✘ | 67.09 |
| | ✔ | **62.89** |
| PGD-2 | ✘ | 50.37 |
| | ✔ | **33.75** |

Table 8: Classification accuracy (%) under different *black-box* query-based attacks on **CIFAR-10**.

| Method | Iterations | PGD-AT | PGD-AT + **HE** | ALP | ALP + **HE** | TRADES | TRADES + **HE** |
|---|---|---|---|---|---|---|---|
| ZOO | - | 73.47 | 74.65 | 72.70 | 74.90 | 71.92 | 74.65 |
| SPSA | 20 | 73.64 | 73.66 | 73.11 | 74.45 | 72.12 | 72.36 |
| | 50 | 68.93 | 69.31 | 68.39 | 68.28 | 68.20 | 68.42 |
| | 80 | 65.67 | 65.97 | 65.14 | 63.89 | 65.32 | 65.62 |
| NES | 20 | 74.68 | 74.87 | 74.41 | 75.91 | 73.40 | 73.47 |
| | 50 | 71.22 | 71.48 | 70.81 | 71.11 | 70.29 | 70.53 |
| | 80 | 69.16 | 69.92 | 68.88 | 68.53 | 68.83 | 69.06 |

**Ablation studies.** To investigate the individual effects caused by the three components FN, WN, and AM in HE, we perform the ablation studies for PGD-AT on CIFAR-10, and attack the trained models with PGD-20. We get the clean and robust accuracy of 86.43% / 54.46% for PGD-AT+FN, and 87.28% / 53.93% for PGD-AT+WN. In contrast, when we apply both the FN and WN, we can get the result of 86.20% / 58.95%. For TRADES, applying appropriate AM can increase $\sim 2\%$ robust accuracy compared to only applying FN and WN. Detailed results are included in the Appendix C.3.

**Adaptive attacks.** A generic PGD attack apply the cross-entropy loss on the model prediction as the adversarial objective, as reported in Table 2. To exclude the potential effect of gradient obfuscation [2], we construct an adaptive version of the PGD attack against our methods, which uses the training loss in Eq. (6) with the scalar $s$ and margin $m$ as the adversarial objective. In this case, when we apply the adaptive PGD-20 / PGD-500 attack, we will get a accuracy of $55.25\%$ / $52.54\%$ for PGD-AT+HE, which is still higher than the accuracy of $53.97\%$ / $51.63\%$ for PGD-AT (quote from Table 2).

**Benchmark attacks.** We evaluate our enhanced models under two stronger benchmark attacks including RayS [11] and AutoAttack [14] on CIFAR-10. We train WRN models via PGD-AT+HE, with weight decay of $5 \times 10^{-4}$ [47]. For RayS, we evaluate on $1,000$ test samples due to the high computation. The results are shown in Table 6, where the trained WRN-34-20 model achieves the state-of-the-art performance (no additional data) according to the reported benchmarks.

### 4.2 Performance under black-box attacks

**Query-based Black-box Attacks.** ZOO [12] proposes to estimate the gradient at each coordinate $e_i$ as $\hat{g}_i$, with a small finite-difference $\sigma$. In experiments, we randomly select one sampled coordinate to perform one update with $\hat{g}_i$, and adopt the C&W optimization mechanism based on the estimated gradient. We set $\sigma$ as $10^{-4}$ and max queries as $20,000$. SPSA [65] and NES [27] can make a full gradient evaluation by drawing random samples and obtaining the corresponding loss values. NES randomly samples from a Gaussian distribution to acquire the direction vectors while SPSA samples from a Rademacher distribution. In experiments, we set the number of random samples $q$ as 128 for every iteration and $\sigma = 0.001$. We show the robustness of different iterations against untargeted score-based ZOO, SPSA, and NES in Table 8, where details on these attacks are in Appendix B.2.

We include detailed experiments on the **transfer-based black-box attacks** in Appendix C.4. As expected, these results show that embedding HE can generally provide promotion under the black-box threat models [9], including the transfer-based and the query-based black-box attacks.

### 4.3 Performance under general-purpose attacks

It has been shown that the adversarially trained models could be vulnerable to rotations [19], image corruptions [20] or affine attacks [62]. Therefore, we evaluate on the benchmarks with distributional shifts: CIFAR-10-C and ImageNet-C [24]. As shown in Table 5, we report the classification accuracy

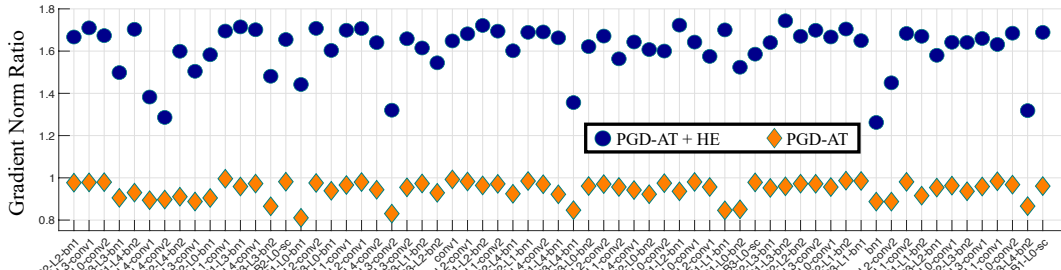

Figure 3: The ratios of $\mathbb{E}(\|\nabla_{\boldsymbol{\omega}}\mathcal{L}(x^*)\|/\|\nabla_{\boldsymbol{\omega}}\mathcal{L}(x)\|)$ w.r.t. different parameters $\omega$, where $x^*$ and $x$ are the adversarial example and its clean counterpart. Higher ratio values indicate more attention on the adversarial examples. 'B' refers to block, 'L' refers to layer, 'conv' refers to convolution.

under each corruption averaged on five levels of severity, where the models are WRN-34-10 trained on CIFAR-10 and ResNet-50 trained on ImageNet, respectively. Here we adopt accuracy as the metric to be consistent with other results, while the reported values can easily convert into the corruption error metric [24]. We can find that our methods lead to better robustness under a wide range of corruptions that are not seen in training, which prevents the models from overfitting to certain attacking patterns.

## 4.4 More empirical analyses

As shown in the left panel of Fig. 2, we separately use PGD-1 to PGD-7 to generate adversarial examples in training, then we evaluate the trained models under the PGD-20 attack. Our method requires fewer iterations of PGD to achieve certain robust accuracy, e.g., applying 3-step PGD in PGD-AT+HE is more robust than applying 7-step PGD in PGD-AT, which largely reduces the necessary computation [69]. To verify the mechanism in Fig. 1, we attack a standardly trained WRN-34-10 (no FN applied) model, applying PGD-1 and PGD-2 with or without FN in the adversarial objective. We report the accuracy in Table 7. As seen, the attacks are more efficient with FN, which suggest that the perturbations are crafted along more effective directions.

Besides, previous studies observe that the adversarial examples against robust models exhibit salient data characteristics [28, 54, 61, 64, 82]. So we visualize the untargeted perturbations on ImageNet, as shown in the right panel of Fig. 2. We can observe that the adversarial perturbations produced for our method have sharper profiles and more concrete details, which are better aligned with human perception. Finally in Fig. 3, we calculate the norm ratios of the loss gradient on the adversarial example to it on the clean example. The model is trained for 70 epochs on CIFAR-10 using PGD-AT. The results verify that our method can prompt the training procedure to assign larger gradients on the crafted adversarial examples, which would benefit robust learning.

## 5 Conclusion

In this paper, we propose to embed the HE mechanism into AT, in order to enhance the robustness of the adversarially trained models. We analyze the intriguing benefits induced by the interaction between AT and HE from several aspects. It is worth clarifying that empirically our HE module has varying degrees of adaptability on combining different AT frameworks, depending on the specific training principles. Still, incorporating the HE mechanism is generally conducive to robust learning and compatible with previous strategies, with little extra computation and simple code implementation.

## Broader Impact

When deploying machine learning methods into the practical systems, the adversarial vulnerability can cause a potential security risk, as well as the negative impact on the crisis of confidence by the public. To this end, this inherent defect raises the requirements for reliable, general, and lightweight strategies to enhance the model robustness against malicious, especially adversarial attacks. In this work, we provide a simple and efficient way to boost the robustness of the adversarially trained models, which contributes to the modules of constructing more reliable systems in different tasks.

## Acknowledgements

This work was supported by the National Key Research and Development Program of China (No.2020AAA0104304), NSFC Projects (Nos. 61620106010, 62076147, U19B2034, U1811461), Beijing Academy of Artificial Intelligence (BAAI), Tsinghua-Huawei Joint Research Program, a grant from Tsinghua Institute for Guo Qiang, Tiangong Institute for Intelligent Computing, and the NVIDIA NVAIL Program with GPU/DGX Acceleration.

## Footnotes

[1]We omit the subscript of $\ell_2$-norm without ambiguity.

[2]Code is available at https://github.com/ShawnXYang/AT_HE.

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
