[Supplementary Material]

# A Proofs

## A.1 Proof of Lemma 1

**Lemma 1.** *Given a loss function $\mathcal{L}$ and under the first-order Taylor expansion, the solution of*

$$\max_{\|x'-x\|_p \leq \epsilon} \mathcal{L}(x')$$

*is $x^* = x + \epsilon \mathbb{U}_p(\nabla \mathcal{L}(x))$. Furthermore, there is $\mathcal{L}(x^*) = \mathcal{L}(x) + \epsilon \|\nabla \mathcal{L}(x)\|_q$, where $\| \cdot \|_q$ is the dual norm of $\| \cdot \|_p$.*

*Proof.* We denote $x' = x + \epsilon v$, where $\|v\|_p \leq 1$. Then we know that $\|x' - x\|_p \leq \epsilon$. Under the first-order Taylor expansion, there is

$$\max_{\|x'-x\|_p \leq \epsilon} \mathcal{L}(x') = \max_{\|v\|_p \leq 1} \left[ \mathcal{L}(x) + \epsilon v^\top \nabla \mathcal{L}(x) \right]$$

$$= \mathcal{L}(x) + \epsilon \max_{\|v\|_p \leq 1} v^\top \nabla \mathcal{L}(x).$$

According to the definition of the dual norm [2], there is $\max_{\|v\|_p \leq 1} v^\top \nabla \mathcal{L}(x) = \|\nabla \mathcal{L}(x)\|_q$, where where $\| \cdot \|_q$ is the dual norm of $\| \cdot \|_p$. Thus we prove that $\mathcal{L}(x^*) = \mathcal{L}(x) + \epsilon \|\nabla \mathcal{L}(x)\|_q$ and $x^* = x + \epsilon \mathbb{U}_p(\nabla \mathcal{L}(x))$. □

## A.2 Proof of Lemma 2

**Lemma 2.** *By derivations, there is*

$$\nabla_{x'} \mathcal{L}_{CE}(f(x'), f(x)) = -\sum_{i \neq j} f(x)_i f(x')_j \nabla_{x'}(W_{ij}^\top z'),$$

*where $W_{ij} = W_i - W_j$, $z' = z(x'; \boldsymbol{\omega})$. When $f(x) = 1_y$, we have $\nabla_{x'} \mathcal{L}_{CE}(f(x'), y) = -\sum_{l \neq y} f(x')_l \nabla_{x'}(W_{yl}^\top z')$.*

*Proof.* By derivations, there is

$$-\nabla_{x'} \mathcal{L}_{\text{CE}}(f(x'), f(x))$$

$$= \nabla_{x'} \left( f(x)^\top \log f(x') \right)$$

$$= \sum_{i \in [L]} f(x)_i \nabla_{x'} \log(f(x')_i)$$

$$= \sum_{i \in [L]} f(x)_i \nabla_{x'} \log \left( \frac{\exp(W_i^\top z')}{\sum_{j \in [L]} \exp(W_j^\top z')} \right)$$

$$= \sum_{i \in [L]} f(x)_i \nabla_{x'} \left( W_i^\top z' - \log \left( \sum_{j \in [L]} \exp(W_j^\top z') \right) \right)$$

$$= \sum_{i \in [L]} f(x)_i \left( \nabla_{x'}(W_i^\top z') - \sum_{j \in [L]} f(x')_j \nabla_{x'}(W_j^\top z') \right)$$

$$= \sum_{i \in [L]} f(x)_i \left( \sum_{j \neq i} f(x')_j \nabla_{x'}(W_{ij}^\top z') \right)$$

$$= \sum_{i \neq j} f(x)_i f(x')_j \nabla_{x'}(W_{ij}^\top z').$$

Specially, when $f(x) = 1_y$, we can obtain based on the above formulas that

$$\nabla_{x'} \mathcal{L}_{\text{CE}}(f(x'), y) = -\sum_{l \neq y} f(x')_l \nabla_{x'}(W_{yl}^\top z').$$

□

# B Related work

In this section, we extensively introduce the related work in the adversarial setting, including the adversarial threat models (Sec. B.1), the adversarial attacks (Sec. B.2), the adversarial training strategy (Sec. B.3), and some recent work on combining metric learning with adversarial training (Sec. B.4).

## B.1 Adversarial threat models

Now we introduce different threat models in the adversarial setting following the suggestions in Carlini et al. [6]. Specifically, a threat model includes a set of assumptions about the adversary's goals, capabilities, and knowledge.

Adversary's goals could be simply fooling the classifiers to misclassify, which is referred to as *untargeted mode*. On the other hand, the goals can be more aggressive to make the model misclassify from a source class into a target class, which is referred to as *targeted mode*.

Adversary's capabilities describe the constraints imposed on the attackers. For the $\ell_p$ bounded threat models, adversarial examples require the perturbation $\delta$ to be bounded by a preset threshold $\epsilon$ under $\ell_p$-norm, i.e., $\|\delta\|_p \leq \epsilon$.

Adversary's knowledge tells what knowledge the adversary is assumed to own. Typically, there are four settings when evaluating a defense method:

- *Oblivious adversaries* are not aware of the existence of the defense $D$ and generate adversarial examples based on the unsecured classification model $F$ [5].
- *White-box adversaries* know the scheme and parameters of $D$, and can design adaptive methods to attack both the model $F$ and the defense $D$ simultaneously [1].
- *Black-box adversaries* have no access to the parameters of the defense $D$ or the model $F$ with varying degrees of black-box access [10].
- *General-purpose adversaries* apply general transformations or corruptions on the images, which are related to traditional research topics on the input invariances [15, 39].

## B.2 Adversarial attacks

Below we show the details of the attack methods that we test on in our experiments. For clarity, we only introduce the untargeted attacks. The descriptions below mainly adopt from Dong et al. [11].

**FGSM** [13] generates an untargeted adversarial example under the $\ell_\infty$ norm as

$$\boldsymbol{x}^{adv} = \boldsymbol{x} + \epsilon \cdot \text{sign}(\nabla_{\boldsymbol{x}} \mathcal{L}_{\text{CE}}(\boldsymbol{x}, y)). \qquad (1)$$

**BIM** [21] extends FGSM by iteratively taking multiple small gradient updates as

$$\boldsymbol{x}_{t+1}^{adv} = \text{clip}_{\boldsymbol{x}, \epsilon}\big(\boldsymbol{x}_t^{adv} + \eta \cdot \text{sign}(\nabla_{\boldsymbol{x}} \mathcal{L}_{\text{CE}}(\boldsymbol{x}_t^{adv}, y))\big), \qquad (2)$$

where $\text{clip}_{\boldsymbol{x}, \epsilon}$ projects the adversarial example to satisfy the $\ell_\infty$ constrain and $\eta$ is the step size.

**PGD** [25] is similar to BIM except that the initial point $\boldsymbol{x}_0^{adv}$ is uniformly sampled from the neighborhood around the clean input $x$, which can cover wider diversity of the adversarial space [34].

**MIM** [10] integrates a momentum term into BIM with the decay factor $\mu = 1.0$ as

$$\boldsymbol{g}_{t+1} = \mu \cdot \boldsymbol{g}_t + \frac{\nabla_{\boldsymbol{x}} \mathcal{L}_{\text{CE}}(\boldsymbol{x}_t^{adv}, y)}{\|\nabla_{\boldsymbol{x}} \mathcal{L}_{\text{CE}}(\boldsymbol{x}_t^{adv}, y)\|_1}, \qquad (3)$$

where the adversarial examples are updated by

$$\boldsymbol{x}_{t+1}^{adv} = \text{clip}_{\boldsymbol{x}, \epsilon}(\boldsymbol{x}_t^{adv} + \alpha \cdot \text{sign}(\boldsymbol{g}_{t+1})). \qquad (4)$$

MIM has good performance as a transfer-based attack, which won the NeurIPS 2017 Adversarial Competition [23]. We set the step size $\eta$ and the number of iterations identical to those in BIM.

**DeepFool** [27] is also an iterative attack method, which generates an adversarial example on the decision boundary of a classifier with the minimum perturbation. We set the maximum number of

iterations as 100 in DeepFool, and it will early stop when the solution at an intermediate iteration is already adversarial.

**C&W** [4] is a powerful optimization-based attack method, which generates an $\ell_2$ adversarial example $\boldsymbol{x}^{adv}$ by solving

$$\underset{\boldsymbol{x}'}{\arg\min}\left\{c\cdot\max(Z(\boldsymbol{x}')_y-\max_{i\neq y}Z(\boldsymbol{x}')_i,0)\right.$$
$$\left.+\|\boldsymbol{x}'-\boldsymbol{x}\|_2^2\right\},\tag{5}$$

where $Z(\boldsymbol{x}')$ is the logit output of the classifier and $c$ is a constant. This optimization problem is solved by an Adam [19] optimizer. $c$ is found by binary search. The C&W attack can also be applied under the $\ell_\infty$ threat model with the adversarial loss function $\max(Z(\boldsymbol{x}')_y-\max_{i\neq y}Z(\boldsymbol{x}')_i,0)$, using the iterative crafting process.

**ZOO** [7] has been proposed to optimize Eq. (5) in the black-box manner through queries. It estimates the gradient at each coordinate as

$$\hat{g}_i=\frac{\mathcal{L}(\boldsymbol{x}+\sigma\boldsymbol{e}_i,y)-\mathcal{L}(\boldsymbol{x}-\sigma\boldsymbol{e}_i,y)}{2\sigma}\approx\frac{\partial\mathcal{L}(\boldsymbol{x},y)}{\partial x_i},\tag{6}$$

where $\mathcal{L}$ is the objective in Eq. (5), $\sigma$ is a small constant, and $\boldsymbol{e}_i$ is the $i$-th unit basis vector. In our experiments, we perform one update with $\hat{g}_i$ at one randomly sampled coordinate. We set $\sigma=10^{-4}$ and max queries as $20,000$.

**NES** [17] and **SPSA** [32] adopt the update rule in Eq. (2) for adversarial example generation. Although the true gradient is unavailable, NES and SPSA give the full gradient estimation as

$$\hat{\boldsymbol{g}}=\frac{1}{q}\sum_{i=1}^{q}\frac{\mathcal{J}(\boldsymbol{x}+\sigma\boldsymbol{u}_i,y)-\mathcal{J}(\boldsymbol{x}-\sigma\boldsymbol{u}_i,y)}{2\sigma}\cdot\boldsymbol{u}_i,\tag{7}$$

where we use $\mathcal{J}(\boldsymbol{x},y)=Z(\boldsymbol{x})_y-\max_{i\neq y}Z(\boldsymbol{x})_i$ instead of the cross-entropy loss, $\{\boldsymbol{u}_i\}_{i=1}^q$ are the random vectors sampled from a Gaussian distribution in NES, and a Rademacher distribution in SPSA. We set $\sigma=0.001$ and $q=128$ in our experiments, as default in the original papers.

### B.3 Adversarial training

Adversarial training (AT) is one of the most effective strategies on defending adversarial attacks, which dominates the winner solutions in recent adversarial defense competitions [3, 23]. The AT strategy stems from the seminal work of Goodfellow et al. [13], where the authors propose to craft adversarial examples with FGSM and augment them into the training data batch in a mixed manner, i.e., each mini-batch of training data consists of a mixture of clean and crafted adversarial samples. However, FGSM-based AT was shown to be vulnerable under multi-step attacks, where Wong et al. [34] later verify that random initialization is critical for the success of FGSM-based AT. Recent work also tries to solve the degeneration problem of one-step AT by adding regularizers [33]. Another well-known AT strategy using the mixed mini-batch manner is ALP [18], which regularizes the distance between the clean logits and the adversarial ones. But later Engstrom et al. [12] successfully evade the models trained by ALP. As to the mixed mini-batch AT, Xie and Yuille [35] show that using an auxiliary batch normalization for the adversarial part in the data batch can improve the performance of the trained models.

Among the proposed AT frameworks, the most popular one is the PGD-AT [25], which formulates the adversarial training procedure as a min-max problem. Zhang et al. [38] propose the TRADES framework to further enhance the model robustness by an additional regularizer between model predictions, which achieves state-of-the-art performance in the adversarial competition of NeurIPS 2018 [3]. However, multi-step AT usually causes high computation burden, where training a robust model on ImageNet requires tens of GPU workers in parallel [22, 36]. To reduce the computational cost, Shafahi et al. [31] propose the FreeAT strategy to reuse the back-propagation result for crafting the next adversarial perturbation, which facilitate training robust models on ImageNet with four GPUs running for two days.

### B.4 Metric learning + adversarial training

Previous work finds that the adversarial attack would cause the internal representation to shift closer to the "false" class [26, 24]. Based on this observation, they propose to introduce an extra triplet loss

term in the training objective to capture the stable metric space representation, formulated as

$$
\begin{aligned}
&\mathcal{L}_{\text{trip}}(z(x^*), z(x_p), z(x_n)) \\
&= [D(z(x^*), z(x_p)) - D(z(x^*), z(x_n)) + \alpha]^+,
\end{aligned}
\tag{8}
$$

where $\alpha > 0$ is a hyperparameter for margin, $x^*$ (anchor example) is an adversarial counterpart based on the clean input $x$, $x_p$ (positive example) is a clean image from the same class of $x$; $x_n$ (negative example) is a clean image from a different class. Here $D(u, v)$ is a distance function. Mao et al. [26] employ an angular distance as $D(u, v) = 1 - \cos \angle(u, v)$; Li et al. [24] apply the $\ell_\infty$ distance as $D(u, v) = \|u - v\|_\infty$. In the implementation, these methods apply some heuristic strategies to sample triplets, in order to alleviate high computation overhead. For example, Mao et al. [26] select the closest sample in a mini-batch as an approximation to the semi-hard negative example. However, the optimization on sampled triplets is still computationally expensive and could introduce class biases on unbalanced datasets [16].

Zhang and Wang [37] apply a feature-scatter solver for the inner maximization problem of AT, which is different from PGD. Instead of crafting each adversarial example based on its clean counterpart, the feature-scatter solver generate the adversarial examples in batch to utilize inter-sample interactions, via maximizing the optimal transport (OT) distance between the clean and adversarial empirical distributions. In the implementation, they use practical OT-solvers to calculate the OT distance and maximize it w.r.t. the adversarial examples. However, the calculation of the OT distance will increase the computational burden for the AT procedure. Besides, the feature-scatter solver also leads to potential threats for the trained models to be evaded by adaptive attacks, e.g., feature attacks, as discussed before[12].

## C  More empirical results

In this section, we provide more empirical results and setups. In our experiments, we apply NVIDIA P100 / 2080Ti GPUs, as well as the Apex package to execute training for FastAT [8, 34]. On CIFAR-10, all the models are trained by four GPUs in parallel for PGD-AT, ALP, and TRADES.

### C.1  Code references

To ensure that our experiments perform fair comparison with previous work, we largely adopt the public codes and make minimal modifications on them to run the trials. Specifically, we refer to the codes of TRADES[3] [38], FreeAT[4] [31], FastAT[5] [34] and the corrupted datasets[6] from Hendrycks and Dietterich [14]. The codes are mostly based on PyTorch [30].

### C.2  Datasets

The CIFAR-10 dataset [20] consists of 60,000 32x32 colour images in 10 classes, with 6,000 images per class. There are 50,000 training images and 10,000 test images. We perform RandomCrop with 4 padding and RandomHorizontalFlip in training as the data augmentation. The ImageNet (ILSVRC 2012) dataset [9] consists of 1.28 million training images and 50,000 validation images in 1,000 classes. As to the data augmentation, we perform RandomResizedCrop and RandomHorizontalFlip in training; Resize and CenterCrop in test. The image size is 256 and the crop size is 224.

### C.3  Extensive ablation studies

Different choices of the scale $s$ and the margin $m$ in HE lead to different trade-offs between the clean accuracy and the adversarial robustness of the trained models, as shown in Table 1. This kind of trade-off is ubiquitous w.r.t. the hyperparameter settings in different AT frameworks [18, 38].

Table 1: Classification accuracy (%) on **CIFAR-10**. The training framework is TRADES + HE with different scale $s$ and margin $m$. We report the performance on clean inputs and under PGD-20 attack.

| Defense | Scale $s$ | Margin $m$ | Clean | PGD-20 |
|---|---|---|---|---|
| | 15 | 0.0 | 82.53 | 60.35 |
| | 15 | 0.1 | **85.00** | 61.13 |
| | 15 | 0.2 | 84.88 | **62.02** |
| | 15 | 0.3 | 82.99 | 61.54 |
| | 15 | 0.4 | 78.05 | 58.05 |
| TRADES + HE | 15 | 0.5 | 74.71 | 56.27 |
| | 1 | 0.2 | **89.34** | 50.33 |
| | 5 | 0.2 | 85.70 | 58.75 |
| | 10 | 0.2 | 85.30 | 60.17 |
| | 15 | 0.2 | 84.88 | **62.02** |
| | 20 | 0.2 | 77.67 | 57.51 |

## C.4   Transfer-based black-box attacks

Due to the adversarial transferability [28, 29], the black-box adversaries can construct adversarial examples based on the substitute models and then feed these examples to evade the original models. In our experiments, we apply PGD-AT, ALP, and TRADES to train the substitute models, respectively. To generate adversarial perturbations, we employ the untargeted PGD-20 [25] and MIM-20 [10] attacks, where the MIM attack won both the targeted and untargeted attacking tracking in the adversarial competition of NeurIPS 2017 [23]. In Fig. 1, we show the results of transfer-based attacks against the defense models trained without or with the HE mechanism. As expected, we can see that applying HE can also better defend transfer-based attacks.

Figure 1: Classification accuracy (%) under the *black-box* transfer-based attacks on **CIFAR-10**. The substitute models are PGD-AT, ALP and TRADES separately. * indicates white-box cases.

## C.5   Full results of m-HE on CIFAR-10

In Table 2, we evaluate the white-box performance of the combinations of the modified HE (m-HE) with PGD-AT, ALP, and TRADES. We set the parameters with $s = 15$ and $m = 0.1$. We can see that m-HE is more effective than HE when combining with PGD-AT, FreeAT and Fast AT that exclusively train on adversarial examples. In contrast, HE performs better than m-HE when combining with the frameworks training on the mixture of clean and adversarial examples, e.g., ALP and TRADES.

Table 2: Classification accuracy (%) on **CIFAR-10** under the *white-box* threat model. The perturbation $\epsilon = 0.031$, step size $\eta = 0.003$, following the setting in Zhang et al. [38].

| Defense | Clean | PGD-20 | PGD-500 | MIM-20 | FGSM | DeepFool | C&W-$\ell_\infty$ |
|---|---|---|---|---|---|---|---|
| PGD-AT | **86.75** | 53.97 | 51.63 | 55.08 | 59.70 | 57.26 | 84.00 |
| PGD-AT + **HE** | 86.19 | 59.36 | 57.59 | 60.19 | **63.77** | **61.56** | **84.07** |
| PGD-AT + **m-HE** | 86.25 | **59.90** | **58.46** | **60.50** | 63.70 | 59.47 | 83.71 |
| ALP | 87.18 | 52.29 | 50.13 | 53.35 | 58.99 | 59.40 | 84.96 |
| ALP + **HE** | **89.91** | **57.69** | 51.78 | **58.63** | **65.08** | **65.19** | **87.86** |
| ALP + **m-HE** | 89.23 | 57.09 | **53.34** | 58.04 | 63.81 | 60.74 | 87.21 |
| TRADES | 84.62 | 56.48 | 54.84 | 57.14 | 61.02 | **60.70** | 81.13 |
| TRADES + **HE** | **84.88** | **62.02** | **60.75** | **62.71** | **65.69** | 60.48 | **81.44** |
| TRADES + **m-HE** | 84.30 | 61.83 | 60.43 | 62.67 | 65.49 | 60.51 | 80.53 |

## C.6 Full results on CIFAR-10-C and ImageNet-C

In Table 3 and Table 4 we provide the full classification accuracy results of different defenses on CIFAR-10-C and ImageNet-C [14], respectively. These reports include detailed accuracy under 75 combinations of severity and corruption.

Table 3: Classification accuracy (%) on **CIFAR-10-C**. Full results on different combination of severity and corruption. Here 'S' refers to the severity from 1 to 5, 'P' refers to PGD-AT, 'A' refers to ALP, 'T' refers to TRADES.

| Defense | S | Noise | | | Blur | | | | Weather | | | | Digital | | | |
|---|---|---|---|---|---|---|---|---|---|---|---|---|---|---|---|---|
| | | Gauss | Shot | Impulse | Defocus | Glass | Motion | Zoom | Snow | Frost | Fog | Bright | Contra | Elastic | Pixel | JPEG |
| P | 1 | 85.97 | 86.3 | 83.55 | 86.15 | 81.92 | 83.65 | 82.86 | 86.12 | 84.88 | 84.27 | 87.15 | 82.37 | 82.06 | 86.23 | 85.17 |
| | 2 | 84.16 | 85.82 | 79.92 | 84.8 | 82.06 | 80.09 | 82.49 | 84.4 | 80.84 | 76.22 | 86.68 | 59.77 | 82.2 | 85.62 | 84.64 |
| | 3 | 81.22 | 83.08 | 76.79 | 82.87 | 81.55 | 76.32 | 81.12 | 82.39 | 75.24 | 65.71 | 85.7 | 40.64 | 81.13 | 85.33 | 84.41 |
| | 4 | 79.4 | 81.2 | 69.55 | 80.67 | 76.18 | 76.39 | 80.17 | 78.03 | 75.93 | 51.96 | 83.87 | 23.09 | 79.65 | 84.27 | 83.86 |
| | 5 | 77.48 | 78.02 | 62.8 | 74.7 | 76.72 | 71.66 | 77.77 | 75.69 | 72.9 | 30.37 | 76.86 | 16.9 | 78.95 | 82.37 | 83.69 |
| P+**HE** | 1 | 85.37 | 85.67 | 83.7 | 85.63 | 81.51 | 83.75 | 82.67 | 85.33 | 84.26 | 84.1 | 86.45 | 82.43 | 81.69 | 85.43 | 85.05 |
| | 2 | 83.69 | 84.99 | 80.97 | 84.5 | 81.76 | 80.66 | 82.33 | 83.99 | 80.33 | 76.84 | 86.0 | 61.36 | 81.92 | 85.15 | 84.52 |
| | 3 | 81.14 | 82.46 | 78.01 | 82.65 | 81.2 | 76.73 | 81.11 | 81.43 | 75.71 | 66.59 | 85.17 | 41.76 | 81.2 | 84.63 | 84.05 |
| | 4 | 79.74 | 81.25 | 71.67 | 80.83 | 76.14 | 77.03 | 80.09 | 77.77 | 76.35 | 52.75 | 83.31 | 23.65 | 80.01 | 83.77 | 83.67 |
| | 5 | 77.76 | 77.99 | 66.37 | 75.58 | 76.67 | 72.68 | 78.16 | 75.35 | 73.28 | 31.96 | 77.44 | 16.36 | 78.61 | 81.84 | 83.23 |
| A | 1 | 86.57 | 86.93 | 84.13 | 86.53 | 83.04 | 84.12 | 83.07 | 86.4 | 85.77 | 85.12 | 87.4 | 83.21 | 82.18 | 86.46 | 85.79 |
| | 2 | 84.96 | 86.31 | 80.48 | 85.01 | 82.65 | 81.04 | 82.73 | 84.86 | 82.19 | 77.27 | 87.24 | 61.34 | 82.11 | 86.02 | 85.09 |
| | 3 | 81.84 | 83.55 | 77.23 | 82.84 | 81.83 | 76.91 | 81.15 | 82.76 | 76.57 | 65.68 | 86.36 | 41.96 | 81.32 | 85.73 | 84.63 |
| | 4 | 80.05 | 81.64 | 69.53 | 80.65 | 77.1 | 77.09 | 80.17 | 78.56 | 77.54 | 51.09 | 84.74 | 24.92 | 80.13 | 84.46 | 84.23 |
| | 5 | 78.01 | 78.6 | 62.97 | 74.69 | 76.96 | 71.99 | 77.75 | 76.13 | 74.26 | 28.39 | 78.66 | 17.85 | 78.81 | 82.79 | 83.68 |
| A+**HE** | 1 | 88.61 | 89.27 | 85.56 | 89.58 | 83.58 | 87.05 | 86.36 | 88.71 | 88.62 | 88.2 | 90.1 | 86.55 | 85.93 | 89.37 | 88.52 |
| | 2 | 85.89 | 88.25 | 81.13 | 88.15 | 83.79 | 83.83 | 85.88 | 87.0 | 86.2 | 82.37 | 89.9 | 69.33 | 85.97 | 88.72 | 87.62 |
| | 3 | 81.61 | 83.99 | 76.29 | 86.14 | 83.69 | 79.81 | 84.68 | 85.71 | 82.47 | 73.58 | 89.13 | 51.2 | 84.94 | 87.98 | 86.99 |
| | 4 | 78.72 | 81.42 | 67.61 | 84.08 | 75.73 | 80.34 | 83.73 | 82.52 | 82.79 | 61.5 | 88.06 | 30.06 | 83.36 | 86.92 | 86.89 |
| | 5 | 76.26 | 76.64 | 60.84 | 78.21 | 77.56 | 75.28 | 81.5 | 81.76 | 79.36 | 38.48 | 84.46 | 16.58 | 82.01 | 84.19 | 86.37 |
| T | 1 | 83.74 | 84.16 | 81.61 | 84.01 | 79.97 | 81.98 | 80.69 | 84.16 | 83.57 | 82.53 | 85.21 | 79.91 | 79.47 | 84.22 | 83.27 |
| | 2 | 81.84 | 83.44 | 78.34 | 82.61 | 79.85 | 78.62 | 80.04 | 82.96 | 79.7 | 74.42 | 84.78 | 57.63 | 79.8 | 83.45 | 82.64 |
| | 3 | 78.63 | 80.56 | 74.84 | 80.58 | 79.55 | 75.09 | 78.9 | 80.79 | 73.97 | 63.06 | 83.78 | 39.34 | 79.05 | 83.07 | 82.32 |
| | 4 | 77.09 | 78.42 | 67.85 | 78.62 | 74.7 | 75.0 | 77.9 | 76.33 | 75.08 | 49.91 | 82.13 | 24.6 | 77.48 | 82.2 | 81.83 |
| | 5 | 74.8 | 75.27 | 61.88 | 73.4 | 74.55 | 71.03 | 75.78 | 73.39 | 72.41 | 28.5 | 74.41 | 17.54 | 76.86 | 80.31 | 81.53 |
| T+**HE** | 1 | 83.06 | 83.78 | 81.12 | 83.96 | 79.42 | 82.13 | 81.28 | 83.69 | 83.22 | 82.31 | 85.0 | 80.2 | 80.35 | 83.84 | 83.24 |
| | 2 | 81.18 | 83.02 | 78.41 | 82.89 | 79.75 | 79.39 | 81.03 | 82.23 | 79.21 | 75.12 | 84.93 | 60.38 | 80.14 | 83.23 | 82.64 |
| | 3 | 78.43 | 80.11 | 75.4 | 81.35 | 79.53 | 76.19 | 79.9 | 80.13 | 73.93 | 65.6 | 83.87 | 41.97 | 79.88 | 83.12 | 82.37 |
| | 4 | 76.85 | 78.63 | 69.59 | 79.78 | 74.03 | 76.62 | 78.95 | 76.34 | 74.51 | 52.36 | 82.2 | 25.59 | 78.33 | 82.16 | 82.07 |
| | 5 | 74.89 | 75.55 | 64.55 | 74.77 | 75.32 | 71.98 | 76.92 | 73.74 | 71.79 | 31.56 | 75.64 | 17.07 | 77.74 | 80.17 | 81.66 |

Table 4: Classification accuracy (%) on **ImageNet-C**. Full results on different combination of severity and corruption. Here 'S' refers to the severity from 1 to 5, 'F' refers to FreeAT.

| Defense | S | Noise | | | Blur | | | | Weather | | | | Digital | | | |
|---|---|---|---|---|---|---|---|---|---|---|---|---|---|---|---|---|
| | | Gauss | Shot | Impulse | Defocus | Glass | Motion | Zoom | Snow | Frost | Fog | Bright | Contra | Elastic | Pixel | JPEG |
| F | 1 | 54.26 | 53.23 | 44.00 | 33.91 | 42.25 | 43.97 | 38.64 | 42.57 | 45.83 | 11.00 | 56.42 | 18.29 | 48.45 | 53.91 | 54.74 |
| | 2 | 45.84 | 43.08 | 32.60 | 26.68 | 34.06 | 34.70 | 33.36 | 26.16 | 28.50 | 4.41 | 53.20 | 6.59 | 31.78 | 52.98 | 53.92 |
| | 3 | 29.74 | 28.36 | 23.48 | 16.82 | 25.22 | 23.83 | 26.93 | 22.69 | 17.22 | 1.54 | 47.98 | 1.38 | 49.64 | 50.19 | 53.34 |
| | 4 | 12.96 | 10.66 | 8.91 | 11.10 | 19.35 | 15.20 | 23.52 | 11.98 | 15.51 | 1.20 | 39.56 | 0.40 | 46.09 | 45.24 | 51.57 |
| | 5 | 3.28 | 4.85 | 2.73 | 7.25 | 12.29 | 11.05 | 18.84 | 11.75 | 10.28 | 0.43 | 28.73 | 0.34 | 32.82 | 41.55 | 49.19 |
| F+**HE** | 1 | 55.14 | 53.27 | 44.29 | 38.15 | 46.08 | 47.57 | 41.90 | 45.92 | 50.32 | 15.25 | 58.73 | 23.31 | 51.15 | 56.17 | 57.03 |
| | 2 | 43.96 | 40.02 | 29.08 | 30.34 | 37.97 | 38.39 | 36.21 | 30.21 | 34.26 | 6.59 | 56.47 | 9.23 | 34.61 | 55.40 | 56.23 |
| | 3 | 24.49 | 23.14 | 18.38 | 18.52 | 27.98 | 26.44 | 29.38 | 26.83 | 22.34 | 2.39 | 52.39 | 1.71 | 52.64 | 52.65 | 55.77 |
| | 4 | 9.37 | 8.37 | 5.89 | 11.71 | 21.45 | 16.42 | 25.61 | 15.22 | 20.43 | 1.85 | 45.56 | 0.47 | 48.94 | 47.82 | 54.17 |
| | 5 | 2.23 | 3.95 | 1.63 | 7.09 | 12.93 | 11.59 | 20.68 | 14.89 | 14.36 | 0.58 | 36.21 | 0.41 | 36.25 | 43.81 | 52.04 |

## Footnotes

[1]https://github.com/Line290/FeatureAttack

[2]https://openreview.net/forum?id=Syejj0NYvr&noteId=rkeBhuBMjS

[3]https://github.com/yaodongyu/TRADES

[4]https://github.com/mahyarnajibi/FreeAdversarialTraining

[5]https://github.com/locuslab/fast_adversarial

[6]https://github.com/hendrycks/robustness