[Reviews · NeurIPS 2020]

Review 1

Summary and Contributions: This paper proposes the idea of enhancing the adversarial training framework with Hyper-spherical Embedding. In particular, the paper uses two normalization techniques to encourage the model to focus only on the angular information. With a negligible additional cost, the proposed method significantly improves the models robustness against a variety of adversarial attacks. - Update after discussion period: I read the rebuttal and the concerns of other reviewers. The rebuttal addressed my main concerns. I agree with other reviewers on the fact that most of the improvements are rather minor, but that, still, these improvements were strongly consistent across several dimensions (threat models, datasets, hyperparameters, and adv. training schemes). Given these considerations, I vote for acceptance.

Strengths: * This paper proposes a novel idea of generating more powerful adversarial attacks during adversarial training frameworks. These attacks correlate more with the angular information in the feature space to eliminate possible biases caused from withe the norms of the norm of the weights in the last linear layer, or the norm of the feature representation. * The paper is well motivated with a nice, simple and neat theoretical analysis. * The proposed method is demonstrated to work well with extensive experimental results on different adversarial training frameworks, two datasets, and a variety of threat models.

Weaknesses: Despite the interesting theoretical and experimental results, there are several caveats. * In lines (131-139) the paper claims that the adversarial perturbations generated with feature normalization technique are more effective. This claim was never confirmed later. Although models trained adversarially with Hypersphere Embedding (HE) seems to enjoy more robustness characteristics, this could be due to very different reasons, other than the aforementioned, e.g. the learnt representation is more clustered under HE. To demonstrate such a claim, an experiment like the following would be required: generate adversarial attacks against models trained with HE (e.g. TRADES + HE) to fool models trained without HE (e.g. TRADES). If the attack success rate is larger with such attacks, then the claim is confirmed. * It would be sufficient to replace Figure 1 with an actual synthetic experiment on small 2-D dataset to confirm. * The idea of feature normalization for better learning on hard adversarial examples is elegant. However, even normalizing the values to the range [-1,1] might not solve the problem. Still, the larger values will have larger contribution, right? The paper lacks some discussion on that aspect. * The modified version of HE in section 3.5 does not co-op with the idea of the paper. The dependence on the value of the angle rather than the cosine will alleviate what was mentioned earlier about the weak and strong examples. Moreover, the method seems to have negligible improvement only with adversarial training, and not with either ALP nor TRADES (Table 2 in the appendix). The paper does not elaborate on the idea behind such phenomenon. Perhaps moving the section with these experiments (along with results) to the appendix would suffice. * The paper reports the results against adaptive attacks when HE was combined with adversarial training only. The paper would improve by reporting more extensive results studying this aspect (similar to Table 2). * It is necessary to have at least a brief discussion about the related work in the main paper.

Correctness: Yes, methodology is appropriate for assessing robustness of the proposed method.

Clarity: Yes, the paper is well written, well structured and easy to follow. Including some related work within the text (not only in the supplementary) could enhance the reading.

Relation to Prior Work: Yes, however the paper elaborated on that in the appendix. A brief discussion in the main paper is necessary upon acceptance.

Reproducibility: Yes

Additional Feedback: In addition to the raised points in the weaknesses, there are few small aspects that need further clarification/adjustment: * The choice of the hyper-parameters (s, m) seems to have significant effect (up to 12 points). Is there any heuristic for reasonable choice for these hyper-parameters? * Table 5 in the main paper is hard to read. Highlight the more robust models with bold in a similar fashion to Tables 2, 3 and 4. Similar comment applies to Tables 4 and 5 in the appendix. * L39: resulted? or resulting? * Reference 32 is taken from arXiv. I think the official publication is already available.


Review 2

Summary and Contributions: Summary: This paper introduced a new adversarial training method by incorporating the hypersphere embedding (HE) mechanism. The authors claimed that the proposed method is lightweight yet effective the analyses showed that HE is well coupled to benefit the robustness of the adversarially trained models. Various experiments on several datasets verified the effectiveness of the proposed method.

Strengths: Strength: 1 . A new adversarial training method with little computational overhead 2 . Comes with intuitive explanations 3 . Contains extensive experiments

Weaknesses: Weakness: 1 . Manipulating the logits could be dangerous 2 . Marginal improvement considering the worst attack 3 . Black-box attacks not effective

Correctness: The claims are generally correct

Clarity: The paper is in good shape and well written

Relation to Prior Work: Yes

Reproducibility: Yes

Additional Feedback: 1 . The proposed method focused on improving the linear layer/logits output of the current state-of-the-art robust training methods by combining weight normalization, feature normalization and angular margins together. Note that all the changes are focused on the last linear layer of the classifier. The good thing is that it has little computational overhead. However, I would say that manipulating the logits could be dangerous in robust training. For example, simply dividing the linear layer’s weight with a large constant may lead to large improvement in PGD robust accuracy as it makes it harder for gradient back-propgation. But this does not really lead to better robust models. The authors did not do any extreme operations like this (which is good), however, it still could lead to a false sense of improvements, which may require stronger evaluations to test it out. 2 . Following the previous point, I am glad that the authors test their proposed method with various attacks. However, it still does not fully clear my concerns. First, I think model robustness is defined on the worst case (the worst robust accuracy). From this point of view, in Table 2, the improvement on HE is rather marginal (~1% for PGD-AT, ALP and TRADES). Second, for Table 3, 4, as I said in the previous point, manipulating the logits may give you a false sense of improvements over PGD, and therefore should be further verified via other stronger attacks. Finally, It is good to see the authors also include black-box experiments, however, the included black-box attacks are rather weak and it is not really informative. I would suggest the authors try stronger ones that are proven to be effective on attacking robust models: "RayS: A Ray Searching Method for Hard-label Adversarial Attack." arXiv preprint arXiv:2006.12792 (2020). 3 . The so-called “first-order adversary” is actually not something new. The following paper also discussed and derived the similar terms (referred as Linear Minimization Oracle in their paper). "A Frank-Wolfe framework for efficient and effective adversarial attacks." AAAI (2020). As well as in the following paper (referred as optimal perturbation to linear function) "Adversarial Training is a Form of Data-dependent Operator Norm Regularization." arXiv preprint arXiv:1906.01527 (2019). The authors might want to also comment on these works. Also notice that this first-order adversary” only approximates 1-step FGSM actually, which may not be representative for N-step PGD case. Other Comments: 1 . In Table 1, the training objective for TRADES should be KL rather than CE, right? I think the entropy term cannot be neglected. 2 . Why use multiplicative s (instead of additive) to avoid numerical issues? I wonder if there is any intuition behind that. ========================== I have read the authors' response as well as other reviewers' comments. I appreciate the extra experiments conducted by the authors and the response indeed clears most of my concerns here. Therefore, I would like to raise my score. I notice that your reported AutoAttack and RayS numbers are based on PGD+HE instead of your best performaning (in the paper) TRADES + HE model. Is it because PGD+HE actually get better result? If so, I would encourage the authors to provide some comments on this and also apply these careful checks on other experiments as well.


Review 3

Summary and Contributions: This paper incorporates hypersphere embedding mechanism into adversarial training. The proposed algorithm normalizes the features in the penultimate layer and the weights in the softmax layer with an additive angular margin. Justifications and empirical evaluations are provided to show its effectiveness in improving model robustness.

Strengths: The authors provide a couple of justifications on why the proposed hypersphere embedding step will help adversarial training. The experimental results show its effectiveness in a convincing way.

Weaknesses: Although Section 3 provides some explanations on why the proposed method will work which is good, but I found it really hard to follow. There is a lack of coherence of this section. I suggest authors to reorganize this section and highlight the key justifications at the beginning (If possible, clearly states the benefits into a couple of theorems).

Correctness: I didn't check every details, but the intuition and the proposed method make sense to me.

Clarity: The paper is generally well-written.

Relation to Prior Work: There is a lack of sufficient discussions on prior works or comparisons with related literature in the main paper, though a related work section is provided in the supplementary materials. In particular, I can not see clearly how the proposed hypersphere embedding mechanism differs from [1]. [1] Hao Wang, Yitong Wang, Zheng Zhou, Xing Ji, Dihong Gong, Jingchao Zhou, Zhifeng Li, and Wei Liu. Cosface: Large margin cosine loss for deep face recognition.

Reproducibility: Yes

Additional Feedback: The detailed comments and questions are listed as follows: 1. Section 3 is hard to follow. Can you explain or summarize in a high-level way why the proposed hypersphere embedding step can lead to the benefits introduced in line 36-40 in the introduction? 2. For the second claimed benefit 'better learning on hard (adversarial) examples', why is it always helpful to bias the adversarial training to devote more efforts on learning hard examples? How do you define the hardness of an example? What if there are examples that are too hard to robustly learn (for example, training images that are even difficult to recognize by human)? 3. Table 2 provides robust accuracy with respect to different adversarial attack. Compared with baseline methods, the proposed hypersphere embedding step does not always improve robust accuracy (e.g. FAB attack for ALP and TRADES). For a better evaluation, I would recommend the authors to include the robust accuracy comparisons in Table 2 against the union of these attack. ========================== The authors' feedback addressed all my questions, so I would like to raise my overall score. I would recommend to include the simple binary classification example (like in the response) to get across the effects of the proposed hypersphere embedding terms.


Review 4

Summary and Contributions: This work suggests the combination of the hypersphere embedding mechanism with state-of-the-art adversarial training procedures. Theoretical justification is provided in section 3 of the paper and extensive experimental analysis is provided in section 4.

Strengths: Soundness of the claims: the authors provide good theoretical background and justification for the hypersphere embedding mechanism; the theoretical details appear to be correct. The authors also provide extensive empirical evaluation. These results also appear to be correct - the reviewer randomly sampled and read through some of the code. Significance and novelty of the contribution: the experimental results portray significant improvements in several experiments over state-of-the-art adversarial training procedures. While the reviewer is not extremely familiar with the adversarial training literature, the contributions appear to be novel. This also appears to be justified by the extensive literature review provided by the authors. Relevance to NeurIPS: this work is extremely relevant to NeurIPS. Many of the adversarial training procedures referenced by the authors have appeared in NeurIPS previously in some form or fashion.

Weaknesses: The reviewer observes three weaknesses: 1. The reviewer would appreciate more discussion of the role of the margin (m) both in the theory and experiment sections. The reviewer saw the section in Appendix C.3 but would appreciate at least more intuitive justification as the authors do so well in Figure 1 with respect to feature normalization. 2. While the AT-HE often surpasses (sometimes significantly surpasses) base AT procedure, several times it does not as observed in Table 2. The reviewer would appreciate some discussion for why the authors believe this happens. This will be useful for other researchers interested in the hypersphere embedding mechanism. 3. The authors claim several times that the hypersphere embedding module is "lightweight" and "requires little extra computation". While there is some discussion of the training time in section 4.1 and results provided in Table 3 with respect to the FAST-AT procedures, the reviewer would appreciate more consideration of the increase in training time at least in the appendix for completeness.

Correctness: The claims and method appear to be correct. The authors convey good understanding of correct methodology.

Clarity: While there are several typos and a couple grammatical issues in the document, the paper is very well written. The flow of the text is logical, on average good intuition is provided for methods, and related work is referenced well throughout the paper.

Relation to Prior Work: Yes, the authors provide extensive citation of prior work. Relation to prior work is incorporated throughout the document and is effective for the flow of the paper.

Reproducibility: Yes

Additional Feedback: The reviewer suggests addressing the three weaknesses in the appendix of the full version of the paper on arXiv. These will be helpful for boosting intuition for newcomers to the hypersphere embedding mechanism in adversarial training. Also, the reviewer especially liked the visualization in Figure 2.

[Author Response · NeurIPS 2020]

We thank all the reviewers for their valuable comments. Below, we address the detailed comments of each reviewer.

**To Reviewer #1. Perturbations crafted with FN:** The transfer accuracy under PGD-10 for TRADES+HE → TRADES is 63.20%, and for TRADES → TRADES+HE is 65.90%. We also apply PGD-1 and PGD-2 w/wo FN to attack a standard WRN-34-10 model, and report the accuracy in Table A. As seen, the attacks are more efficient with FN. **Choice of (s, m):** Heuristically, the value range of $s$ is based on the averaged logit norms of standard training, which is around 10. The margin range $m$ is chosen to be around $\cos 30° \approx 0.15$.

Table A: PGD w/wo FN.

| Attack | FN | Acc (%) |
|---|---|---|
| PGD-1 | ✖ | 67.09 |
|  | ✔ | **62.89** |
| PGD-2 | ✖ | 50.37 |
|  | ✔ | **33.75** |

**FN on learning hard adversarial examples:** Indeed, larger $\nabla_{\boldsymbol{\omega}} \cos(\theta)$ could have larger contribution in the mini-
batch. However, since $\cos(\theta)$ is bounded, its gradient will be smaller when the sample is gradually well-learned
(i.e., $\cos(\theta) \to 1$). Then other samples that are not well-learned will dynamically contribute more. In contrast, if we
do not apply FN, the unbounded $\|z\|$ will cause vicious circles (i.e., large $\nabla_{\boldsymbol{\omega}} \|z\|$ leads to larger $\|z\|$), and the easy
examples will keep dominating the training. As we show in Table B, our mechanism can help the model to achieve
SOTA performance under the stronger attacks. **Fig. 1, Sec. 3.5, and other comments:** Thank you for the suggestions.
We will construct synthetic demos and better organize Sec. 3.5; We will involve more related work and re-check the
references into published versions; We will include complete results on adaptive attacks and polish our Tables.

**To Reviewer #2. Evaluation under stronger attacks:** We evaluate under two stronger attacks including RayS[1] and AutoAttack[2] on CIFAR-10. We train WRN models via PGD-AT+HE, with weight decay of $5 \times 10^{-4}$. For RayS, we evaluate on $1,000$ test samples due to the high computation. The results are shown in Table B, where the trained WRN-34-20 model achieves SOTA performance (no additional data) according to the reported benchmarks.

Table B: Acc. (%) of PGD-AT+HE.

| Model | Clean | RayS | AA |
|---|---|---|---|
| WRN-34-10 | 86.25 | 57.8 | 53.16 |
| WRN-34-20 | 85.14 | 59.0 | 53.74 |

**First-order adversary:** We cited Simon-Gabriel et al. [55] in line 100 when we introducing first-order adversaries,
and we never claimed it as one of our contributions. Thank you for pointing out other related work and we'll discuss on
them in the revision. **Training objective of TRADES:** In the 4-th line of Sec. 5.2 in TRADES paper [78], the authors
clarify that they choose $\mathcal{L}$ as the cross-entropy loss, so we provide the formula under the cross-entropy loss in Table 1.
In the TRADES code[3], they apply KL loss, and we also use KL loss to keep consistency. **Why use multiplicative**
**scalar:** First, the softmax function is invariant to any additive offset, i.e., $\mathbb{S}(x + s) = \mathbb{S}(x)$. After executing FN and
WN, the logit values will be constrained to $[-1, 1]$, which will make the training loss be trapped at a very high value
and vanish the gradients. Then a multiplicative scalar $s$ can enlarge the value interval and promote the training process.

**To Reviewer #3. High-level intuition of line 36-40:** In the binary classification, the CE objective equals to maximizing
$\mathcal{L}(x) = (W_0 - W_1)^\top z = \|W_{01}\| \|z\| \cos(\theta)$ on an input $x$ with label $y = 0$. **(i)** If $x$ is correctly classified, there is
$\mathcal{L}(x) > 0$, and adversaries aim to craft $x'$ such that $\mathcal{L}(x') < 0$. Since $\|W_{01}\|$ and $\|z\|$ are always positive, they cannot
alter the sign of $\mathcal{L}$. Thus FN and WN encourage the adversaries to attack the crucial component $\cos(\theta)$; **(ii)** In a data
batch, points with larger $\|z\|$ will dominate (vicious circle on increasing $\|z\|$), which makes the model ignore the critical
component $\cos(\theta)$. FN alleviates this problem, and well-learned hard examples will dynamically have smaller weights
during training since $\cos(\theta)$ is bounded; **(iii)** When there are much more samples of label 0, the CE objective will tend
to have $\|W_0\| \gg \|W_1\|$ to minimize the loss. WN can relieve this trend and encourage $W_0$ and $W_1$ to diversify in
directions; **(iv)** The role of margin is analogous to it in SVM. We will better reorganize Sec. 3 in the revision.

**Relation to Cosface:** The HE mechanism in Eq. (6) has the same form as Cosface [65], and we contribute to applying
it in the adversarial training with both theoretical and empirical analyses. We will detail the relation in the revision.
**Hard examples:** We define the hardness w.r.t. $\nabla_{\boldsymbol{\omega}} \mathcal{L}(x)$, where hard (adversarial) examples usually correspond to the
worst cases for a model. As you suggest, bias towards them may be unreasonable in the sense of human perception.
However, under a threat model (e.g., $8/255$, $\ell_\infty$) in which we evaluate our defenses, the ground-truth labels are assumed
to be invariant and always well-defined. **Union of attacks:** The improvement on PGD-AT is more significant (SOTA as
in Table B) since its framework is better aligned with our analyses. We will provide more comparisons in the revision.

**To Reviewer #5.** Thank you for your kind words. **The role of margin:** The margin $m$ encourages a larger gap between the logit of the true label and other logits. This makes the learned features more aligned with the corresponding softmax weights, as well as more distinguished weight directions. We will detail on the margin with more empirical results. **Results in Table 2:** We can observe that the HE mechanism is better suitable for PGD-AT, since its framework is more consistent with our analyses. Our newest experiment results in Table B demonstrate the SOTA performance of PGD-AT+HE. In contrast, encoding HE into ALP and TRADES is less formally justified, and we will try to elaborate on them with fine-tuned formulas in the revision. **Training time:** We show the training time in Table C on CIFAR-10, and we can see that HE only introduces little extra computation.

Table C: Average training time (minutes) per epoch.

| Method | Time |
|---|---|
| PGD-AT | 19.22 |
| PGD-AT+HE | 19.23 |
| ALP | 20.89 |
| ALP+HE | 20.95 |
| TRADES | 25.83 |
| TRADES+HE | 25.84 |

Reference: [1]RayS benchmark: github.com/uclaml/RayS. [2]AutoAttack benchmark: github.com/fra31/auto-attack. [3]TRADES code: github.com/yaodongyu/TRADES.


[Meta-Review · NeurIPS 2020]

We thank the authors for their careful response which, along with reviewer discussion, cleared up many concerns. Reviewers still have some concerns that we hope the authors will address in their paper (see reviews). Overall though, the reviewers felt this was an interesting method with an appealing computational cost.